# School absenteeism among children and adolescents aged 6–19 years with sickle cell disease in Uganda: A comparative cross-sectional study

Juliane Peninah Nattimba[1]*, Sabrina Bakeera-Kitaka[1], Joseph Rujumba[1], Ruth Namazzi[1], Ivan Segawa[2], Kizza Lubega[3], Alphonse Taban[4], Edison Mworozi[1], Phillip Kasirye[1], Deogratias Munube[1], Grace Ndeezi[1], Sarah Kiguli[1], Thereza Piloya[1]

1 Department of Pediatrics and Child Health, Makerere University, Kampala, Uganda, 2 Department of Clinical Epidemiology Unit, Makerere University, Kampala, Uganda, 3 Makerere University Joint AIDS Program, Kampala, Uganda, 4 Department of Orthopedics, Makerere University, Kampala, Uganda

* nattimbapenina@gmail.com

## Abstract

### Introduction

Sickle cell disease (SCD) contributes substantially to school absenteeism due to recurrent pain, infections, and frequent hospital visits that interrupt learning and long-term academic progress. In Uganda, where the SCD burden is high, the extent and drivers of absenteeism are not well documented, limiting the development of supportive school health strategies. This study assessed the prevalence and associated factors of absenteeism among children and adolescents with SCD to inform education and health policy.

### Methods

A comparative cross-sectional study was conducted at Mulago National Referral Hospital's Sickle Cell Clinic in Kampala, Uganda, from 18 July to 03 November 2024. Children aged 6–19 years with SCD and their siblings or peers were consecutively enrolled. Chronic absenteeism: missing ≥10% of expected school days in the previous term (~89 days), was the primary outcome. Sociodemographic, school-related, and clinical data were collected through structured interviews and medical record review. Chi-square and Wilcoxon rank-sum tests were used to compare participant characteristics, while logistic regression identified factors associated with chronic absenteeism, generating odds ratios (ORs) and 95% confidence intervals (CIs).

### Results

A total of 358 participants were enrolled (179 with SCD and 179 peers). Children with SCD missed a median of 5 school days per term (interquartile range [IQR]: 2–10),

---

which permits unrestricted use, distribution, and reproduction in any medium, provided the original author and source are credited.

**Data availability statement:** The data described in this article can be freely and openly accessed at the Figshare repository (Nattimba, Juliane Peninah (2026). School absenteeism and associated factors among children and adolescents attending sickle cell clinic, Mulago National Referral hospital, Kampala district. figshare. Dataset. https://doi.org/10.6084/m9.figshare.31290772.v1).

**Funding:** This study was supported by the National Heart, Lung, and Blood Institute of the National Institutes of Health under Award Number D43TW012466. The content is solely the responsibility of the authors and does not necessarily represent the official views of the National Institutes of Health. There was no additional external funding received for this study. The funders had no role in study design, data collection and analysis, decision to publish, or preparation of the manuscript.

**Competing interests:** The authors have declared that no competing interests exist.

compared with 0 days (IQR: 0–3) among peers (p < 0.001). Chronic absenteeism affected 33% of children with SCD versus 6% of peers (OR 7.3, 95% CI: 3.7–14.4). In multivariable analysis, class repetition (adjusted OR [aOR] 2.5, 95% CI: 1.1–5.9) and hospital admissions (aOR 3.6, 95% CI: 1.9–6.8) were significantly associated with absenteeism.

## Conclusions

Learners with SCD experienced significantly high rates of absenteeism compared to their peers, with one-third chronically absent. Frequent hospital admissions and prior class repetition were strongly associated, highlighting the need for integrated health-education strategies to enhance attendance and academic outcomes.

## Introduction

Sickle Cell Disease (SCD) is a group of inherited blood disorders that impair oxygen transport, resulting in chronic anemia, recurrent pain crises, and fatigue, factors that disrupt daily activities such as school attendance and learning [1]. One of the most measurable consequences of this burden is school absenteeism, which disrupts learning and can contribute to long-term educational disadvantage [2]. School performance correlated positively with hemoglobin concentration (r = 0.26, *p* < 0.001), suggesting lower hemoglobin levels are associated with poorer academic outcomes among children with SCD [3]. These findings illustrate how the physiological burden of SCD can directly translate into educational disparities, making school absenteeism an important marker of the disease's wider social impact.

Globally, 300,000–400,000 babies are born with SCD each year, with sub-Saharan Africa (SSA) accounting for over 75% of this burden [4,5]. In Uganda alone, approximately 15,000 children are born with SCD annually, with a national prevalence of 13.3% and a sickle cell anemia prevalence of 0.8% [6,7]. Historically, 50–80% of affected children in SSA died before the age of five [5]. Survivors often experience recurrent pain episodes, infections, end-organ damage, and delayed growth and development, all of which limit educational achievement and perpetuate cycles of poverty [5]. Although interventions such as neonatal screening, hydroxyurea therapy, prophylactic treatments, and health education have improved survival [7,8], these have not adequately addressed the long-term educational and socio-economic challenges faced by children with SCD [7].

School absenteeism among children with SCD is well-documented in high-income countries. For example, in the USA, students with SCD miss an estimated 20–40 school days annually due to painful vaso-occlusive crises, hospitalizations, routine clinic visits, infections, and fatigue [2]. These absences contribute to significant learning gaps, poorer academic performance, and higher risks of dropout [2]. In low-resource settings like Uganda, these challenges are magnified by poverty, transportation difficulties, limited access to healthcare, and rigid school systems that do not accommodate chronic illnesses [9]. While Uganda's Ministry of Education

promotes daily attendance tracking and has introduced electronic data systems, these measures rarely account for the unique needs of children with chronic diseases, such as SCD [10]. Consequently, children with SCD often remain inadequately supported in school, contributing to educational vulnerability recognized globally by UNICEF and UNESCO [11].

Despite recognition of these challenges, few targeted interventions exist to help children with SCD remain engaged in school, especially in low-income settings. Most existing studies on absenteeism focus on high-income countries or address chronic illnesses in general, rather than the specific clinical and psychosocial context of SCD in sub-Saharan Africa [3, 12]. In Uganda, there is limited evidence on the prevalence and associated factors of school absenteeism among children with SCD [13], and national education policies do not distinguish between illness-related absenteeism from other causes. This evidence gap constrains the design of appropriate school health policies and support systems. This study, therefore, aimed to assess the prevalence and associated factors of school absenteeism among children and adolescents with SCD attending a sickle cell clinic in Uganda. The findings were expected to inform inclusive education policies, guide targeted interventions, and improve educational outcomes for children living with chronic health conditions.

## Materials and methods

### Study design and setting

We conducted a comparative cross-sectional study at the Sickle Cell Clinic (SCC) of Mulago National Referral Hospital in Kampala, Uganda. This was a comparative cross-sectional study conducted among children aged 6–19 years. Participants were recruited based on sickle cell disease (SCD) status rather than absenteeism status. Children with SCD were consecutively enrolled during routine clinic visits at the national referral hospital's Sickle Cell Clinic. For each enrolled child with SCD, a sibling or age-comparable peer without SCD was recruited from the same household or community. Chronic absenteeism was assessed at the time of interview based on school attendance during the previous term. Participants were not selected based on absenteeism status, and no retrospective identification of cases and controls was performed. As the largest public and national referral hospital in the country, Mulago offers free services to over 480,000 children and adolescents with SCD annually. The SCC operates Monday to Friday under the Department of Pediatrics and registers more than 70 new patients each month, with daily attendance ranging from 40 to 70 children and adolescents. It is the largest specialized SCD clinic in Uganda, functioning both as a referral center and as a primary healthcare facility for neighboring communities as well as for patients referred from across the country. In addition, SCD services are available at 13 regional referral hospitals and about 80 smaller satellite clinics nationwide.

### Participants

The study recruited children and adolescents with SCD, along with their caregivers, and siblings or peers as a comparison group. Children and adolescents with SCD were eligible if they were aged 6–19 years, had a confirmed diagnosis of SCD documented in their medical records, attended the SCC during the study period, were enrolled in school, and were able to understand English or Luganda (the dominant local language in Central Uganda). The comparison group comprised siblings or peers from the same household who were aged 6–19 years, within 3–4 years of age of the child with SCD, enrolled in school, and able to understand English or Luganda. We excluded children and adolescents with SCD who were severely ill, as well as siblings with chronic illnesses (such as renal, heart, or liver disease, including SCD, or with reported symptoms) or those who were ill during the study period.

### Study variables

The primary outcome was chronic school absenteeism, defined as missing ≥10% of expected school days in a term, consistent with UNICEF criteria [11]. In Uganda, a school term typically lasts 12–13 weeks (~89 instructional days), excluding

weekends and official school holidays [14]. This percentage was calculated by dividing the number of days missed by the total number of expected school days. School days missed were recorded as an integer based on caregiver or participant self-report for the previous school term.

Sociodemographic variables included age in years (retrieved from medical records or reported by caregivers) and sex at birth (female or male). School-related characteristics included current class, highest educational level attained (nursery, primary, secondary, or tertiary), whether the child had ever repeated a class (yes/no), school type (public or private; day or boarding), and reported reasons for missing school. Caregivers provided information on their relationship to the child, marital status, biological parent availability (both alive and living at home), family size and number of children with SCD, education level, employment status, monthly household income (reported in Uganda shillings [UGX] and converted to US dollars [USD]; 1 UGX = 0.00027 USD), and whether the household income was sufficient to pay school fees on time (yes/no).

Medical history among children with SCD included the number and duration of clinic visits, hospital admissions, episodes of painful crises, and blood transfusions. Clinical characteristics retrieved from clinic records included haemoglobin genotype, current medications, child's status at the time of data collection (outpatient or hospitalized), history of stroke (yes/no), and medical conditions reported in the previous school term.

## Study procedures

We employed a consecutive sampling approach, between 18th July 2024 and 3rd November 2024, enrolling all eligible children and adolescents until the required sample size of 179 was reached. For comparison, we recruited 179 siblings or peers without SCD, related as closely as possible by age, to minimize potential confounding by socioeconomic, cultural, or demographic factors. Children with SCD were first identified during clinic visits or through SCC records. For each participant with SCD, the sibling closest in age was selected as the comparator. If no eligible sibling was available, a peer living in the same household and closest in age was chosen. When several potential peers or siblings were available, one was randomly selected. Caregivers confirmed the SCD status of siblings or peers; those with unknown status but without symptoms were included. Participants without siblings or peers were still eligible for inclusion.

Recruitment occurred during routine clinic visits or by contacting families ahead of their scheduled clinic appointments. Eligibility was confirmed at enrollment using SCC records and caregiver reports. Caregivers were requested to bring the identified sibling or peer to the next clinic visit.

Data were collected by three trained medical staff using a pretested structured questionnaire, administered in either English or Luganda, the two most widely spoken languages at the clinic. Interviews were conducted face-to-face and lasted about 45 minutes. The questionnaire, which had been refined through pre-testing with 10 participants (five with SCD and five siblings/peers, later excluded from the main dataset), captured demographic and school-related information, medical history, and physical examination findings.

School absenteeism was measured by asking caregivers and participants to report the number of school days missed during the previous term. Whenever possible, this information was cross-checked using pupils' books and clinic medical records. Data quality was ensured through daily reviews of completed questionnaires, double data entry, and logic checks. Equipment used during data collection was calibrated and operated by qualified personnel.

Families received a reimbursement of 20,000 Ugandan shillings (~US$5.40) after completing the questionnaire.

Completed questionnaires were immediately reviewed for completeness and accuracy. Participants were assigned unique ID codes, and their names were not collected. No identifiable information was available to the researchers. Data was entered into the Research Electronic Data Capture (REDCap) platform hosted at Global Health Uganda. De-identified data was stored on a password-protected computer and backed up on a secure online repository. The final dataset was cleaned and prepared for analysis.

## Data analysis

With chronic school absenteeism as the primary outcome, the required sample size was calculated to detect a difference in the proportion of chronic absenteeism between children with sickle cell disease (SCD) and their siblings or peers without SCD. The calculation was based on a two-proportion comparison formula. We assumed an expected prevalence of chronic absenteeism of 38% among children with SCD ($p_1 = 0.38$) and 19% among siblings/peers ($p_2 = 0.19$). The assumed proportions ($p_1$ and $p_2$) were informed by prior studies reporting school absenteeism among children with SCD and comparison groups in similar contexts [12]. A 5% level of significance ($Z\alpha = 1.96$) and 80% power ($Z\beta = 0.84$) were applied. The resulting minimum sample size was 179 participants per group [12].

Numerical variables (e.g., age) were summarized as medians with interquartile ranges (IQR) and compared between groups using the Wilcoxon rank-sum (Mann–Whitney U) test. Categorical variables (e.g., sex at birth) were summarized as proportions and compared using chi-square tests. Absenteeism (missing at least one school day) was analyzed both as a count and as a proportion. Chronic absenteeism was categorized as <10% for no chronic absenteeism or ≥10% for chronic absenteeism, in line with the UNICEF threshold.

Logistic regression was used to identify factors associated with chronic absenteeism. Bivariate analysis generated crude odds ratios (OR) and their 95% confidence intervals (CI). Variables with $p < 0.25$ in bivariate analysis were considered for inclusion in the multivariable model. The final models retained variables with $p < 0.05$ and their confounders, using standard techniques for assessing interaction and confounding (10% cut-off).

## Ethical considerations

Ethical approval was obtained from the Makerere University School of Medicine Research and Ethics Committee (Mak-SOMREC-2024–925). Administrative clearance was obtained from the administration of Mulago National Referral Hospital and the in-charge of the SCC. Informed written consent and assent were obtained from each participant before the start of the interview sessions. The research procedures were thoroughly explained to all participants, and their understanding and comprehension were assessed in a language they understood. Participants provided written consent or assent as appropriate. For children younger than 8 years, only caregiver consent was obtained, while for those aged 8 years and above, both caregiver consent and the child's assent were obtained in accordance with the national research ethics guidelines [15].

## Results

A total of 302 families were approached or contacted by phone to participate in the study at the SCC. Data collection began on 18th July 2024 and concluded on 3rd November 2024, after screening 482 children. Of these, 123 children with SCD and one sibling/peer were excluded, leaving 179 children with SCD and 179 siblings/peers (141 siblings and 38 peers) included in the study and analysis. Exclusions were due to dropping out of school (31), being older than 19 years (67), parents discontinuing school fees due to illness (22), or declining assent/consent (4).

Children with SCD had a median age of 11 years (IQR 9–13), and nearly half were female (50%, 89/179) (Table 1). Their siblings/peers had a similar median age of 11 years (IQR 9–14), with a slightly higher proportion of females (58%, 103/179). Although most children in both groups were in primary school, a higher proportion of siblings/peers attended secondary school (24%, 42/179) compared with children with SCD (15%, 27/179). Class repetition was significantly higher among children with SCD (24%, 42/177) than among siblings/peers (14%, 25/179, $p = 0.018$).

Overall, 88.8% (159/179) of children with SCD missed at least one school day compared to 36.9% (66/179) of siblings/peers. The median number of missed school days was 5 (IQR 2–10) for children with SCD, compared to 0 (IQR 0–3) for siblings/peers ($p < 0.001$). The main reason for absenteeism among children with SCD was complications related to SCD (60%, 108/179), whereas acute illnesses accounted for most absenteeism among siblings/peers (18%, 33/179) (Table 1).

**Table 1. Sociodemographic and school characteristics of all the 358 children and adolescents that participated in the study.**

| Characteristic | With SCD (N = 179), n (%) or Median (IQR) | Sibling/peer (N = 179), n (%) or Median (IQR) |
|---|---|---|
| **Sex** | | |
| Male | 90 (50.3) | 76 (42.5) |
| Female | 89 (49.7) | 103 (57.5) |
| **Current age** | | |
| Median (IQR), years | 11(9 –13) | 11(9 –14) |
| **Same biological parents as child with SCD** | | |
| No | | 38 (21.2) |
| Yes | | 141 (78.8) |
| **Highest level of education** | | |
| Nursery | 19 (10.6) | 13 (7.3) |
| Primary | 133 (74.3) | 123 (68.7) |
| Secondary | 27 (15.1) | 42 (23.5) |
| Tertiary | 0 (0.0) | 1 (0.5) |
| **Day or Boarding School** | | |
| Day | 158 (88.3) | 127 (71.0) |
| Boarding | 21 (11.7) | 52 (29.0) |
| **Private or public school** | | |
| Private | 137 (76.3) | 144 (80.5) |
| Public/Government | 42 (23.5) | 35 (19.5) |
| **Ever repeated class** | | |
| No | 135 (76.3) | 154 (86.0) |
| Yes | 42 (23.7) | 25 (14.0) |

A total of 179 caregivers participated, most of whom were mothers or female guardians (88%, 158/179), with a median age of 38 years (IQR 32–45) (S1 Table). Families had a median of four children (IQR 3–6), including a median of one child with SCD (IQR 1–2). The median household income was 81 USD (IQR 54–162); 74% (132/179) reported that it was insufficient to pay school fees on time, and 72% (129/179) reported that their children missed school due to financial constraints. Among the parents of children with SCD, 52% (78/149) of fathers and 50% (84/169) of mothers had attained secondary education (S2 Table).

Table 2 presents the clinical characteristics of children with SCD and explores their association with school absenteeism. Clinical records showed that all children with SCD (100%, 179/179) were taking folic acid, and 92% (172/179) were on hydroxyurea (Table 2). In the previous school term, children with SCD had a median of one clinic visit (range 0–5), no hospital admissions (range 0–5), one pain crisis (range 0–10), and no blood transfusions (range 0–5). Among those admitted, the median length of hospital stay was 7 days (range 1–60). Most had one clinic visit (44%, 79/179), no admissions (73%, 131/179), no pain crises (31%, 55/179), and no blood transfusions (84%, 150/179).

Chronic school absenteeism was reported by 33% (95% CI: 26–40%, 59/179) of children with SCD compared to 6% (95% CI: 3–11%, 11/179) among their siblings/peers (Table 3). Among all 358 children combined, chronic absenteeism was significantly associated with SCD status (OR 7.3, 95% CI: 3.7–14.4, $p < 0.001$), school type (boarding vs. day school; OR 0.3, 95% CI: 0.1–0.8, $p = 0.004$), and class repetition (OR 2.6, 95% CI: 1.4–4.7, $p = 0.003$). None of the examined variables confounded the relationship between SCD status and chronic absenteeism in multivariable analysis.

Table 2. Clinical characteristics of children with sickle cell disease and their association with school absenteeism during the previous term (n = 179).

| Characteristic | n (%) or Median (IQR) |
|---|---|
| **Number of visits to SCC,** Median (IQR) | 1 (1-2) |
| **Routine or scheduled visits,** Median (IQR) | 1 (0-1) |
| **Total number of times the child was admitted,** Median (IQR) | 0 (0-1) |
| **Total number of days admitted,** Median (IQR) | 7 (3-10) |
| **Total number of episodes of pain crises,** Median (IQR) | 1 (0-2) |
| **Number of blood transfusions,** Median (IQR) | 0 (0-0) |
| **Medication prescribed**<br>Folic acid<br>Malaria prophylaxis<br>Anti-malarials<br>Hydroxy urea<br>Pneumonia prophylaxis<br>Antibiotics<br>Analgesics<br>Morphine<br>Oral Rehydrating Solution<br>Salbutamol | 179 (100.0)<br>172 (96.1)<br>6 (3.4)<br>165 (92.2)<br>2 (1.1)<br>39 (21.8)<br>119 (66.5)<br>9 (5.0)<br>1 (0.6)<br>1 (0.6) |
| **Status of the child's clinic attendance**<br>Daycare (OPD)<br>Hospitalized | 171 (95.5)<br>8 (4.5) |
| **Reason for hospitalization**<br>Vaso-occlusive crisis<br>SCD with gastritis | 7 (87.5)<br>1 (12.5) |
| **Conditions in the last term**<br>Pneumonia<br>Acute splenic sequestration<br>Mild infection without fever/significant fever, or URTI<br>Vaso-occlusive crisis<br>New stroke<br>Leg ulcer<br>Other | 3 (1.7)<br>1 (0.6)<br>44 (24.6)<br>62 (34.6)<br>13 (7.3)<br>1 (0.6)<br>61 (34.1) |

## Discussion

We aimed to assess the extent of school absenteeism and its associated factors among 358 children and adolescents with SCD. Chronic absenteeism (denoting ≥10% of school days missed) was reported in 33% of children with SCD compared with 6% of their peers. Chronic absenteeism was significantly associated with SCD status, school type, and class repetition. Within the SCD group, absenteeism was associated with class repetition and the number of hospital admissions.

About one-third of children with SCD were chronically absent from school, missing a median of five days per term and being seven times more likely to miss school than their peers. The findings of this study contribute to the growing global discourse on the educational implications of sickle cell disease (SCD). Across both high-income countries and low- and middle-income countries, children living with SCD consistently experience higher levels of school absenteeism compared to their peers [1,2,16]. However, the magnitude and consequences of absenteeism are shaped by contextual factors, including healthcare quality and access, availability of school-based, and broader social support systems [2].

**Table 3. Factors associated with chronic school absenteeism among all 358 children and adolescents.**

| Characteristic | Chronic absenteeism, Yes, n (%) | OR (95% CI) | p value |
|---|---|---|---|
| SCD status | | | |
| With SCD | 59 (33.0) | 7.3 (3.7-14.4) | **<0.001** |
| Sibling/peer | 11 (6.1) | Reference | |
| Gender | | | |
| Female | 32 (16.7) | 0.7 (0.4-1.1) | 0.139 |
| Male | 38 (22.9) | Reference | |
| Age, Mean (±SD), years | 11.0 (±3.3) | 1.0 (0.9-1.1) | 0.878 |
| Highest education | | | |
| Secondary or higher | 11 (15.7) | 0.7 (0.4-1.5) | 0.368 |
| Primary or lower | 59 (20.5) | Reference | |
| School type | | | |
| Boarding | 6 (8.2) | 0.3 (0.1-0.8) | **0.004** |
| Day | 64 (22.5) | Reference | |
| School type | | | |
| Public/Government | 18 (23.4) | 1.3 (0.7-2.4) | 0.421 |
| Private | 52 (18.5) | Reference | |
| Ever repeated class | | | |
| Yes | 22 (32.8) | 2.6 (1.4-4.7) | **0.003** |
| No | 46 (15.9) | Reference | |

Note. SCD = Sickle Cell Disease, SD = Standard Deviation

Among children with SCD, chronic absenteeism was more likely in those who had ever repeated a class, had frequent SCC visits, hospital admissions, pain crises, or a history of blood transfusions in bivariate analysis (all p < 0.05, (Table 4) In multivariable analysis, chronic absenteeism was independently associated with class repetition (adjusted odds ratio [aOR] 2.5, 95% CI: 1.1–5.9, p = 0.035) and the number of hospital admissions (aOR 3.6, 95% CI: 1.9–6.8, p < 0.001).

In high-income settings, structured educational support services, individualized learning plans, and access to remote learning options may mitigate some of the academic disruption associated with recurrent illness and hospitalization [1]. In contrast, in many resource-limited settings, including Uganda, limited access to compensatory academic support and fewer school-based health accommodations may amplify the reported cases of absenteeism among children with SCD in Africa. The prevalence of chronic absenteeism observed in our study exceeds the reported national absenteeism rates for primary and secondary school students in Uganda (23–24%) [17,18].

Our findings are consistent with evidence from other settings. For example, studies conducted in Nigeria reported that children with sickle cell disease missed an average of six school days per term compared to two days among their peers without the condition [3,19]. Similarly, research from Yemen found that 60% of children with sickle cell disease experienced chronic absenteeism, missing more than 20 school days annually [3,19]. Frequent pain crises, infections, and hospitalizations disrupt school attendance, often requiring additional days for recovery [3]. Low household income (~$81/ month), coupled with the high costs of SCD care, likely compounds absenteeism by delaying school fee payments [20,21].

Consequently, chronic absenteeism may contribute to poor academic performance, disengagement, and increased risk for school dropout [3,21]. In our study, fewer children with SCD advanced to secondary school, suggesting cumulative effects of SCD and absenteeism. Targeted interventions such as remedial classes, financial support, and teacher sensitization could help mitigate these challenges [21]. Schools should recognize SCD as a chronic condition requiring flexible attendance policies and, where feasible, improve access to healthcare in school settings. Aligning clinic visits outside school days may reduce learning disruptions [8,22].

**Table 4. Factors independently associated with chronic school absenteeism among the 179 children and adolescents with SCD.**

| Characteristic | Chronic absenteeism, Yes, n (%) | OR (95% CI) | *P* value | aOR (95% CI) | *p* value |
|---|---|---|---|---|---|
| **Gender** | | | | | |
| Female | 24 (27.0) | 0.6 (0.3-1.1) | 0.091 | – | – |
| Male | 35 (38.9) | Reference | | | |
| **School type** | | | | | |
| Boarding | 4 (19.1) | 0.4 (0.1-1.4) | 0.158 | – | – |
| Day | 55 (34.8) | Reference | | | |
| **Ever repeated class** | | | | | |
| Yes | 20 (47.6) | 2.4 (1.2-4.9) | **0.016** | 2.5 (1.1-5.9) | **0.035** |
| **Household income sufficient to pay school fees on time** | | | | | |
| Yes | 12 (25.5) | 0.6 (0.2-1.3) | 0.209 | – | – |
| **Children missing school due to the failure to pay fees** | | | | | |
| Yes | 47 (36.4) | 1.8 (0.9-3.8) | 0.115 | – | – |
| **Mother's highest level of education** | | | | | |
| ≥Secondary | 31 (27.4) | 0.6 (0.3-1.1) | 0.120 | – | – |
| ≤Primary | 22 (39.3) | Reference | | | |
| **Number of visits to the SCC** | | | | | |
| Mean (±SD) | 1.9 (±1.5) | 1.7 (1.3-2.2) | **<0.001** | 1.3 (1.0-1.8) | 0.076 |
| **Total number of times the child was admitted** | | | | | |
| Mean (±SD) | 0.9 (±0.8) | 3.9 (2.2-6.9) | **<0.001** | 3.6 (1.9-6.8) | **<0.001** |
| **Total number of episodes of pain crises** | | | | | |
| Mean (±SD) | 2.2 (±2.3) | 1.3 (1.1-1.5) | **0.005** | – | – |
| **Number of previous blood transfusions** | | | | | |
| Mean (±SD) | 0.4 (±0.8) | 2.6 (1.3-5.0) | **0.005** | – | – |

Note. SCD = Sickle Cell Disease, SCC = Sickle Cell Clinic, SD = Standard Deviation.

Among children with SCD, class repetition increased the odds of chronic school absenteeism by 2.5 times compared to those making normal progress. Similarly, children with SCD and class repetition in Yemen had a significantly higher school absence: they missed a median of 52 days compared to 14 days among those with no class repetition [16,19,23]. In our study, class repetition is not only an outcome of SCD but also a reinforcer of school absenteeism. From previous studies, we know that children with SCD are likely to repeat at least one grade (class) [19]. Retention is a result of poor academic performance linked with health-related absences, neurocognitive impacts of SCD, and low family income [16,23,24]. In turn, absenteeism itself may increase the risk of class repetition [16]. Retained students with SCD may become demotivated or experience increased social stigma and lowered self-esteem [25,26]. In addition, students with SCD are at highest risk of school drop-out and should be closely monitored and supported by the school system [16].

Our findings indicate that for every additional hospital admission, a child with SCD is nearly four times as likely to chronically miss school compared to those with fewer hospitalizations. Eaton and colleagues found that children with SCD who were frequently hospitalized were absent from school nearly twice as often as those who were rarely admitted [27]. Access to online or virtual schooling remains limited in Uganda. During periods of illness or hospitalization, most children lack structured remote learning or tutoring options, which may intensify the educational consequences of school absenteeism among learners with SCD.

Although hospitalization was rare in our study, those who were admitted lost nearly a week of precious learning time, which makes it difficult to catch up with peers. Recurrent hospital visits and admissions reflect the impact of disease

severity on consistent school participation [7]. Clinical severity, reflected through frequent painful crises, hospitalizations, blood transfusions, and SCD-related complications, is associated with absenteeism, worse school experiences, and learning difficulties [19,28]. Hence, students with SCD would benefit from regular checkups, prophylactic medicines and vaccines, and psychosocial support to prevent severe disease [2,22].

These findings have important implications for clinicians caring for children and adolescents with sickle cell disease. Routine clinical care should include screening for school absenteeism as part of holistic patient assessment, since frequent absence may signal worsening disease severity, psychosocial distress, or barriers to treatment adherence. Early identification and management of pain crises, infections, and other complications that lead to hospitalization may help reduce school disruption. Clinicians should also counsel caregivers on the importance of sustained school attendance and work with families to schedule routine clinic visits, where possible, outside critical school periods. Strengthening psychosocial support and referral systems for learners with repeated absenteeism may further improve both health and educational outcomes.

The findings also have important implications for school administrators and education authorities. School administrators should recognize sickle cell disease as a chronic health condition requiring flexible and supportive attendance policies rather than punitive responses to repeated absence. Improved communication between schools, caregivers, and healthcare providers may help schools better understand illness-related absenteeism and support timely academic catch-up. Teacher sensitization on the challenges faced by learners with SCD is important to reduce stigma, improve classroom support, and encourage retention in school. Where feasible, schools should strengthen school health programs and provide reasonable accommodations such as flexible assessment schedules, remedial learning support, and referral pathways for children experiencing frequent illness-related interruptions in learning.

This study had several strengths, including the use of a comparative design that allowed for direct comparison between children with SCD and their siblings and peers. It also took a comprehensive approach, considering health and academic factors using standardized tools. Nonetheless, limitations exist. First, the cross-sectional design limits the ability to establish temporal relationships between associated factors and chronic absenteeism; therefore, causal inferences cannot be made. Second, school absenteeism was assessed using caregiver or participant self-report for the previous school term, which may be subject to recall bias and misclassification. Caregivers of children with sickle cell disease may recall illness-related absences more readily than caregivers of comparison participants, potentially resulting in differential recall.

Third, as with many observational studies, residual confounding from unmeasured variables cannot be excluded despite multivariable adjustment. Factors such as school-level support systems, teacher attitudes, and household socioeconomic dynamics that were not fully captured in this study may influence patterns of absenteeism. In addition, most participants were recruited from urban and peri-urban areas within the catchment of the national referral hospital. Transportation barriers were not directly assessed; however, such challenges may be more pronounced in rural settings and could further contribute to school absenteeism among children living with chronic illnesses such as SCD.

Finally, participants were recruited from a clinical setting and included only children currently enrolled in school. Children with SCD who had permanently withdrawn from school were not captured, which may have led to underestimation of the true extent of absenteeism among the children with SCD at the Mulago Hospital's Sickle Cell Clinic.

## Conclusions

Children and adolescents with sickle cell disease experience a substantially higher burden of chronic school absenteeism compared to their siblings or peers without the condition. In this study, chronic absenteeism was common among learners with SCD and was associated with hospital admissions and prior class repetition, reflecting the intersection of health-related disruptions and academic vulnerability.

Although causal inferences cannot be drawn from this cross-sectional analysis, the findings contribute to the growing global evidence that school absenteeism is a key educational challenge for children living with SCD. The results highlight

the importance of strengthening collaboration between healthcare providers and educational systems, particularly in resource-limited settings, to better support sustained school participation for learners affected by chronic illness. Future longitudinal studies are needed to clarify temporal relationships and to inform the development of effective, context-sensitive interventions.

## Supporting information

**S1 Table. Socio-demographics for caregivers of children with Sickle cell disease included in the study.** This is the S1 Fig legend.
(DOCX)

**S2 Table. Socio-economic status of the biological parents of the children with Sickle cell disease included in the study.** This is the S2 Fig legend.
(DOCX)

## Acknowledgments

We thank the research participants and clinical staff at Mulago Hospital's Sickle Cell clinic. In addition, we are grateful to the research assistants for collecting the data. Special thanks to Mirembe Vanessa and Kataike Peace for their commitment and hard work.

## Author contributions

**Conceptualization:** Juliane Peninah Nattimba, Sabrina Bakeera-Kitaka, Ruth Namazzi, Ivan Segawa, Kizza Lubega, Phillip Kasirye, Grace Ndeezi, Sarah Kiguli, Thereza Piloya.

**Data curation:** Juliane Peninah Nattimba, Sabrina Bakeera-Kitaka, Joseph Rujumba, Ruth Namazzi, Ivan Segawa, Kizza Lubega, Alphonse Taban, Edison Mworozi, Phillip Kasirye, Deogratias Munube, Grace Ndeezi, Sarah Kiguli, Thereza Piloya.

**Formal analysis:** Juliane Peninah Nattimba, Ivan Segawa.

**Funding acquisition:** Juliane Peninah Nattimba, Sabrina Bakeera-Kitaka, Joseph Rujumba, Ruth Namazzi, Phillip Kasirye, Deogratias Munube, Grace Ndeezi, Sarah Kiguli, Thereza Piloya.

**Investigation:** Juliane Peninah Nattimba, Sabrina Bakeera-Kitaka, Joseph Rujumba, Ruth Namazzi, Ivan Segawa, Kizza Lubega, Alphonse Taban, Edison Mworozi, Phillip Kasirye, Deogratias Munube, Grace Ndeezi, Thereza Piloya.

**Methodology:** Juliane Peninah Nattimba, Sabrina Bakeera-Kitaka, Joseph Rujumba, Ruth Namazzi, Ivan Segawa, Kizza Lubega, Alphonse Taban, Edison Mworozi, Deogratias Munube, Grace Ndeezi, Sarah Kiguli, Thereza Piloya.

**Project administration:** Juliane Peninah Nattimba, Sabrina Bakeera-Kitaka, Joseph Rujumba, Ruth Namazzi, Ivan Segawa, Kizza Lubega, Alphonse Taban, Edison Mworozi, Phillip Kasirye, Deogratias Munube, Grace Ndeezi, Sarah Kiguli, Thereza Piloya.

**Resources:** Juliane Peninah Nattimba, Sabrina Bakeera-Kitaka, Joseph Rujumba, Ruth Namazzi, Phillip Kasirye, Deogratias Munube, Grace Ndeezi, Sarah Kiguli, Thereza Piloya.

**Software:** Juliane Peninah Nattimba, Sabrina Bakeera-Kitaka, Joseph Rujumba, Ruth Namazzi, Ivan Segawa, Sarah Kiguli, Thereza Piloya.

**Supervision:** Juliane Peninah Nattimba, Sabrina Bakeera-Kitaka, Joseph Rujumba, Ruth Namazzi, Kizza Lubega, Alphonse Taban, Edison Mworozi, Phillip Kasirye, Grace Ndeezi, Sarah Kiguli, Thereza Piloya.

**Validation:** Juliane Peninah Nattimba, Sabrina Bakeera-Kitaka, Joseph Rujumba, Ruth Namazzi, Ivan Segawa, Alphonse Taban, Edison Mworozi, Phillip Kasirye, Deogratias Munube, Grace Ndeezi, Sarah Kiguli, Thereza Piloya.

**Visualization:** Juliane Peninah Nattimba, Sabrina Bakeera-Kitaka, Joseph Rujumba, Ruth Namazzi, Ivan Segawa, Kizza Lubega, Alphonse Taban, Edison Mworozi, Phillip Kasirye, Deogratias Munube, Grace Ndeezi, Sarah Kiguli, Thereza Piloya.

**Writing – original draft:** Juliane Peninah Nattimba, Kizza Lubega, Alphonse Taban, Thereza Piloya.

**Writing – review & editing:** Juliane Peninah Nattimba, Sabrina Bakeera-Kitaka, Joseph Rujumba, Ruth Namazzi, Ivan Segawa, Kizza Lubega, Alphonse Taban, Edison Mworozi, Phillip Kasirye, Deogratias Munube, Grace Ndeezi, Sarah Kiguli, Thereza Piloya.

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
