## [Decision Letter · Decision Letter 0]

14 Jan 2026

PONE-D-25-63605School absenteeism among children and adolescents aged 6-19 years with sickle cell disease in Uganda: a comparative cross-sectional studyPLOS One

Dear Dr. Nattimba,

Thank you for submitting your manuscript to PLOS ONE. After careful consideration, we feel that it has merit but does not fully meet PLOS ONE’s publication criteria as it currently stands. Therefore, we invite you to submit a revised version of the manuscript that addresses the points raised during the review process.

Please revise the paper following guidance from the reviewers. In addition, re-check the design which you call a cross-sectional sectional design right from the title. You might have to re-consider giving it the appropriate name as a matched case control study right from the title because this is what you did. The authors should highlight key limitations of a case control study which readers should be aware of. The discussion, try to show how your work adds to the global discourse around SCD beyond the local study setting. Lastly, the entire still needs proof-reading to correct obvious language errors. ==============================

We look forward to receiving your revised manuscript.

Kind regards,

Aloysius Gonzaga Mubuuke

Academic Editor

PLOS One

**Journal Requirements:**

1. When submitting your revision, we need you to address these additional requirements. Please ensure that your manuscript meets PLOS ONE's style requirements, including those for file naming. The PLOS ONE style templates can be found at https://journals.plos.org/plosone/s/file?id=wjVg/PLOSOne_formatting_sample_main_body.pdf and https://journals.plos.org/plosone/s/file?id=ba62/PLOSOne_formatting_sample_title_authors_affiliations.pdf 2. Thank you for stating in your Funding Statement: This study was supported by the National Heart, Lung, and Blood Institute of the National Institutes of Health under Award Number D43TW012466. The content is solely the responsibility of the authors and does not necessarily represent the official views of the National Institutes of Health.  Please provide an amended statement that declares *all* the funding or sources of support (whether external or internal to your organization) received during this study, as detailed online in our guide for authors at http://journals.plos.org/plosone/s/submit-now. Please also include the statement “There was no additional external funding received for this study.” in your updated Funding Statement. Please include your amended Funding Statement within your cover letter. We will change the online submission form on your behalf. 3. Thank you for stating the following financial disclosure: This study was supported by the National Heart, Lung, and Blood Institute of the National Institutes of Health under Award Number D43TW012466. The content is solely the responsibility of the authors and does not necessarily represent the official views of the National Institutes of Health.   Please state what role the funders took in the study.  If the funders had no role, please state: "The funders had no role in study design, data collection and analysis, decision to publish, or preparation of the manuscript." If this statement is not correct you must amend it as needed. Please include this amended Role of Funder statement in your cover letter; we will change the online submission form on your behalf. 4. Thank you for stating the following in the Acknowledgments Section of your manuscript: I thank the research participants and clinical staff at the Sickle Cell Clinic at Mulago Hospital. I am grateful to the research assistants for collecting the data. Special thanks to Mirembe Vanessa and Kataike Peace for their commitment and hard work. I also appreciate the ENRICH program for the guidance and financial resources throughout this journey. We note that you have provided funding information that is not currently declared in your Funding Statement. However, funding information should not appear in the Acknowledgments section or other areas of your manuscript. We will only publish funding information present in the Funding Statement section of the online submission form. Please remove any funding-related text from the manuscript and let us know how you would like to update your Funding Statement. Currently, your Funding Statement reads as follows: This study was supported by the National Heart, Lung, and Blood Institute of the National Institutes of Health under Award Number D43TW012466. The content is solely the responsibility of the authors and does not necessarily represent the official views of the National Institutes of Health.  Please include your amended statements within your cover letter; we will change the online submission form on your behalf. 5. Thank you for uploading your study's underlying data set. Unfortunately, the repository you have noted in your Data Availability statement does not qualify as an acceptable data repository according to PLOS's standards. At this time, please upload the minimal data set necessary to replicate your study's findings to a stable, public repository (such as figshare or Dryad) and provide us with the relevant URLs, DOIs, or accession numbers that may be used to access these data. For a list of recommended repositories and additional information on PLOS standards for data deposition, please see https://journals.plos.org/plosone/s/recommended-repositories. 6. Your ethics statement should only appear in the Methods section of your manuscript. If your ethics statement is written in any section besides the Methods, please delete it from any other section. 7. We notice that your supplementary tables are included in the manuscript file. Please remove them and upload them with the file type 'Supporting Information'. Please ensure that each Supporting Information file has a legend listed in the manuscript after the references list. 8. If the reviewer comments include a recommendation to cite specific previously published works, please review and evaluate these publications to determine whether they are relevant and should be cited. There is no requirement to cite these works unless the editor has indicated otherwise.

**Additional Editor Comments:**

Please re-check the design which you call a cross-sectional sectional design right from the title. You might have to re-consider giving it the appropriate name as a matched case control study right from the title because this is what you did. The authors should highlight key limitations of a case control study which readers should be aware of. The discussion, try to show how your work adds to the global discourse around SCD beyond the local study setting. Lastly, the entire still needs proof-reading to correct obvious language errors.

Reviewers' comments:

Reviewer's Responses to Questions

**Comments to the Author**

1. Is the manuscript technically sound, and do the data support the conclusions?

Reviewer #1: Yes

Reviewer #2: Partly

2. Has the statistical analysis been performed appropriately and rigorously? 

Reviewer #1: Yes

Reviewer #2: Yes

3. Have the authors made all data underlying the findings in their manuscript fully available?

Reviewer #1: Yes

Reviewer #2: Yes

4. Is the manuscript presented in an intelligible fashion and written in standard English?

Reviewer #1: Yes

Reviewer #2: Yes

5. Review Comments to the Author

**Reviewer #1:** This is a well written manuscript describing a cross-sectional study of school absenteeism in school-aged children with SCD compared to age-matched siblings or peers without SCD. The recruitment and data collection are appropriate and well described. The results are clearly presented in text and Tables. The statistical approach is appropriate. The conclusions and limitations suggested appropriately reflect the study results. Literature from both high and low-middle income countries are provided for comparison.

Questions for the authors:

1. please define a school "term" and it's typical duration.

2. The need for or lack thereof for transportation to school can often be a barrier-was this a potential concern in this region or population? Any urban vs rural differences?

3. Please elaborate on availability of online/virtual schooling or tutoring services

**Reviewer #2:** The authors performed a cross-sectional comparative observational study on patients with sickle cell disease and matched peers. The aim of interest was the prevalence of chronic school absenteeism and the associated factors.

The manuscript is well written and easy to read and addresses a very important topic.

There are however some important things to be considered for improvement.

1. In the abstract and the manuscript, the authors use the words "predictors" and "associated factors" interchangeably. However, this is not acceptable. Predictors are only measurable in a cohort design only as opposed to associated factors that can be measured cross-sectionally.

2. In essence, this is a matched case control study and should be presented as such right from the title and the body. The SCD participants are the cases and the matched peers are the controls.

3. In the sample size calculation, it is not clear what "q1" and "q2" stand for. Sample size for two proportion comparison study requires only "p1" (Expected proportion of chronic absenteeism in the SCD participants) and "p2" (Expected proportion of chronic absenteeism in the controls).

4. In the same sample size calculation, there is no reference or explanation for how the p1 and p2 were obtained since these were supposed to be predetermined.

5. In the results, table one should strictly contain baseline demographic characteristics and should not contain the "results". Eg number of school days missed, reasons for missing school etc.

6. I see no use of table 2 in the results, which objective is this answering? We need results on the following

- Prevalence of chronic absenteeism in the two groups with their 95%CI and the comparison (prevalence ratio or OR with CI)

-Bi-variate and multivariate logistic regression on the associated factors (And this should be limited to only the SCD group since its entirely exploratory and the study is not powered for a comparison between the cases and controls).

7. Table 4 can be improved significantly, the crude and adjusted odds ratios can be presented concurrently in the same table (Again forget about the comparison as this is not useful), and on the rows, only present the positive (Yes) instead of one row for "Yes" and another row for "no' for the same variable eg only one row for "ever repeated a class?"

8. Limit the discussion and conclusion mostly for what the study is powered for.

9. Limitations: Please report the traditional limitations for case control study, and in your study, the cases were more likely to remember or have records as regards chronic absenteeism.

6. PLOS authors have the option to publish the peer review history of their article (what does this mean?). If published, this will include your full peer review and any attached files.

Reviewer #1: **Yes:** Carlton Dampier MD

Reviewer #2: **Yes:** Conrad K Muzoora

---

## [Author Response · Author response to Decision Letter 1]

28 Feb 2026

Dear Dr. Mubuuke,

On behalf of my co-authors, I hereby submit the responses to the reviewers’ comments for the manuscript titled “School absenteeism among children and adolescents aged 6–19 years with sickle cell disease in Uganda: a comparative cross-sectional study” (Manuscript ID: PONE-D-25-63605). We thank the reviewers for the careful review of our manuscript and for the constructive guidance provided. We have addressed each comment in detail below and revised the manuscript accordingly.

JOURNAL REQUIREMENTS

Journal Requirement 1: PLOS ONE Style and File Naming

Comment: 1. When submitting your revision, we need you to address these additional requirements.

Response: We confirm that the manuscript has been revised to fully comply with PLOS ONE formatting and style requirements, using the official PLOS ONE templates for the main body and title page. File naming conventions have also been updated in accordance with journal guidelines. All submitted files, including the revised manuscript with tracked changes, the clean manuscript, and its supporting information. In addition, we have adhered to the required naming and formatting standards.

Journal Requirement 2: Funding Statement - Completeness

Comment 2. Thank you for stating in your Funding Statement:

This study was supported by the National Heart, Lung, and Blood Institute of the National Institutes of Health under Award Number D43TW012466. The content is solely the responsibility of the authors and does not necessarily represent the official views of the National Institutes of Health.

Response: The Funding Statement has been amended to declare all sources of support and now explicitly includes the statement indicating that no additional external funding was received.

Revised text: Cover letter

“This study was supported by the National Heart, Lung, and Blood Institute of the National Institutes of Health under Award Number D43TW012466. The content is solely the responsibility of the authors and does not necessarily represent the official views of the National Institutes of Health. There was no additional external funding received for this study.”

Journal Requirement 3: Role of the Funder

Comment 3. Thank you for stating the following financial disclosure:

This study was supported by the National Heart, Lung, and Blood Institute of the National Institutes of Health under Award Number D43TW012466. The content is solely the responsibility of the authors and does not necessarily represent the official views of the National Institutes of Health.

Response: We confirm that the funders had no role in the conduct of the study. The Role of the Funder statement has been clarified accordingly.

Revised text: Cover letter

Journal Requirement 4: Funding Information in Acknowledgments

Comment 4. Thank you for stating the following in the Acknowledgments Section of your manuscript:

I thank the research participants and clinical staff at the Sickle Cell Clinic at Mulago Hospital. I am grateful to the research assistants for collecting the data. Special thanks to Mirembe Vanessa and Kataike Peace for their commitment and hard work. I also appreciate the ENRICH program for the guidance and financial resources throughout this journey.

This study was supported by the National Heart, Lung, and Blood Institute of the National Institutes of Health under Award Number D43TW012466. The content is solely the responsibility of the authors and does not necessarily represent the official views of the National Institutes of Health.

Response: We agree and have removed all funding-related text from the Acknowledgments section. The Acknowledgments now include only expressions of gratitude to participants, clinical staff, and research assistants. All funding information is presented exclusively in the Funding Statement, as per PLOS ONE policy.

Revised text: Section: Acknowledgements, pages-21, lines 408-410

“We thank the research participants and clinical staff at Mulago Hospital’s Sicklecell clinic. In addition, we are grateful to the research assistants for collecting the data. Special thanks to Mirembe Vanessa and Kataike Peace for their commitment and hard work.”

Journal Requirement 5: Data Availability and Repository

Comment 5. Thank you for uploading your study's underlying data set. Unfortunately, the repository you have noted in your Data Availability statement does not qualify as an acceptable data repository according to PLOS's standards.

Response: We thank the journal for this guidance. The minimal dataset required to replicate the study findings has now been deposited in a stable, public repository compliant with PLOS ONE standards, Figshare. The Data Availability Statement has been updated to include the correct repository name and DOI/URL.

Revised text: Availability of data and materials and references, pages: 20, lines-390

‘’The data described in this article can be freely and openly accessed at the Figshare repository (Nattimba, Juliane Peninah (2026). School absenteeism and associated factors among children and adolescents attending sickle cell clinic, Mulago National Referral hospital, Kampala district.. figshare. Dataset. https://doi.org/10.6084/m9.figshare.31290772.v1)’’

Journal Requirement 6: Ethics Statement Location

Comment 6. Your ethics statement should only appear in the Methods section of your manuscript. If your ethics statement is written in any section besides the Methods, please delete it from any other section

Response: We confirm that the ethics statement now appears only in the Methods section. Any duplicate ethics-related text appearing elsewhere in the manuscript has been removed.

Journal Requirement 7: Supporting Information Files

Comment 7. We notice that your supplementary tables are included in the manuscript file. Please remove them and upload them with the file type 'Supporting Information'. Please ensure that each Supporting Information file has a legend listed in the manuscript after the references list.

Response: All supplementary tables have been removed from the main manuscript and uploaded as separate Supporting Information files. Each Supporting Information file is now referenced appropriately in the manuscript, with legends listed after the references section.

Journal Requirement 8: Reviewer-Recommended Citations

Comment 8. If the reviewer comments include a recommendation to cite specific previously published works, please review and evaluate these publications to determine whether they are relevant and should be cited. There is no requirement to cite these works unless the editor has indicated otherwise

Response: We have now included an appropriate citation in the Methods section to support the statistical approach used for the sample size calculation. The reference has been added to clarify the methodological basis and improve the transparency and reproducibility of the study design. The data repository from which relevant comparative or contextual data were obtained has also been appropriately referenced. The reviewers recommended no additional citations.

Revised text: Methods- Data analysis- Page8, Lines 184-186, Availability of data and Materials-Page 20, Lines 390-391

“The assumed proportions (p₁ and p_2_) were informed by prior studies reporting school absenteeism among children with SCD and comparison groups in similar contexts. (12)”

EDITORIAL COMMENTS

Editor Comment 1: Please re-check the design which you call a cross-sectional design right from the title. You might have to reconsider giving it the appropriate name as a matched case-control study because this is what you did.

Response: We respectfully maintain that the study is appropriately classified as a comparative cross-sectional study and not a case-control study. We recognize that the inclusion of two groups may resemble a case-control structure; however, the recruitment process and analytical framework align with cross-sectional methodology.

Participants were recruited based on exposure status (presence or absence of sickle cell disease) rather than outcome status (chronic absenteeism). Children with SCD were consecutively enrolled from the Sickle Cell Clinic during routine visits. For each enrolled child with SCD, a sibling or peer without SCD of similar age was recruited as a comparison participant. Importantly, participants were not selected based on whether they had chronic absenteeism, and absenteeism status was unknown at the time of enrollment.

Both exposure (SCD status) and outcome (chronic absenteeism during the previous school term) were assessed concurrently at a single point in time. No retrospective identification of cases and controls based on absenteeism occurred, and no matching or conditional analytical techniques typical of case-control designs were applied.

Revised text: Section: Methods - Study design and setting, Page:5, Lines:95-99

“This was a comparative cross-sectional study conducted among children aged 6–19 years. Participants were recruited based on sickle cell disease (SCD) status rather than absenteeism status. Children with SCD were consecutively enrolled during routine clinic visits at the national referral hospital’s Sickle Cell Clinic. For each enrolled child with SCD, a sibling or age-comparable peer without SCD was recruited from the same household or community. Chronic absenteeism was assessed at the time of the interview based on school attendance during the previous term. Participants were not selected based on absenteeism status, and no retrospective identification of cases and controls was performed.”

Editor Comment 2: The authors should highlight key limitations of a case-control study which readers should be aware of.

Response: We agree with the Editor that clearly articulating study limitations is essential. While this study is cross-sectional, we recognize that several limitations commonly discussed in observational and case-control studies are relevant. We have therefore expanded the Limitations subsection to address potential sources of bias, including recall bias, lack of temporality, and selection bias. In particular, we discuss the possibility of differential recall of school absenteeism among caregivers of children with sickle cell disease, as well as the exclusion of children who may have permanently withdrawn from school.

Revised text: Section: Discussion - Limitations, Pages: 18-19, Lines:344-362

“First, the cross-sectional design limits the ability to establish temporal relationships between associated factors and chronic absenteeism; therefore, causal inferences cannot be made. Second, school absenteeism was assessed using caregiver or participant self-report for the previous school term, which may be subject to recall bias and misclassification. Caregivers of children with sickle cell disease may recall illness-related absences more readily than caregivers of comparison participants, potentially resulting in differential recall. Third, as with many observational studies, residual confounding from unmeasured variables cannot be excluded despite multivariable adjustment. Factors such as school-level support systems, teacher perceptions, or household socioeconomic dynamics not fully captured in the study may influence absenteeism patterns. Finally, participants were recruited from a clinical setting and included only children currently enrolled in school. Children with sickle cell disease who had permanently withdrawn from school were not captured, which may have led to underestimation of the true extent of absenteeism among the children with SCD at the Mulago Hospital Sickle Cell clinic.”

Editor Comment 3: The discussion, try to show how your work adds to the global discourse around sickle cell disease beyond the local study setting.

Response: The Discussion has been substantially strengthened to situate our findings within the broader global literature on sickle cell disease and educational outcomes. We now compare our findings with studies from both high-income and low- and middle-income countries and highlight how structural constraints, such as limited access to remote learning and frequent healthcare utilization, may amplify educational disadvantage in resource-limited settings. We further emphasize that school absenteeism among children with sickle cell disease is a global concern, but one that manifests differently depending on health system capacity, educational infrastructure, and social support mechanisms. This framing enhances the relevance of our findings beyond Uganda.

Revised text: Section: Discussion, Page: 15–16, Lines:283-295

“The findings of this study contribute to the growing global discourse on the educational implications of sickle cell disease (SCD). Across both high-income countries and low- and middle-income countries, children living with SCD consistently experience higher levels of school absenteeism compared to their peers (1,2,22). However, the magnitude and consequences of absenteeism are shaped by contextual factors, including healthcare access, availability of school-based accommodations, and broader social support systems (2). In high-income settings, structured educational support services, individualized learning plans, and access to remote learning options may mitigate some of the academic disruption associated with recurrent illness and hospitalization (1). In contrast, in many resource-limited settings, including Uganda, limited access to compensatory academic support and fewer school-based health accommodations may amplify the afore reported cases of absenteeism among children with SCD in Africa. “

Editor Comment 4: Lastly, the entire manuscript still needs proofreading to correct obvious language errors.

Response: The entire manuscript has undergone thorough proofreading to corre

---

## [Decision Letter · Decision Letter 1]

17 Apr 2026

PONE-D-25-63605R1School absenteeism among children and adolescents aged 6-19 years with sickle cell disease in Uganda: a comparative cross-sectional studyPLOS One

Dear Dr. Nattimba,

Thank you for submitting your manuscript to PLOS ONE. After careful consideration, we feel that it has merit but does not fully meet PLOS ONE’s publication criteria as it currently stands. Therefore, we invite you to submit a revised version of the manuscript that addresses the points raised during the review process.

**ACADEMIC EDITOR:** The paper has significantly improved and is benefit to the readership of PLOS ONE. The discussion has been strengthened. However, before the authors explain the strengths and limitations of the study towards the end of the discussion, they should include some key implications of the findings from this study first to clinicians who take care of these children with SCD and second, to the school administration authorities especially that they deal with these children while in school. ==============================

We look forward to receiving your revised manuscript.

Kind regards,

Aloysius Gonzaga Mubuuke

Academic Editor

PLOS One

Journal Requirements:

Additional Editor Comments :

The paper has significantly improved and is benefit to the readership of PLOS ONE. The discussion has been strengthened. However, before the authors explain the strengths and limitations of the study towards the end of the discussion, they should include some key implications of the findings from this study first to clinicians who take care of these children with SCD and second, to the school administration authorities especially that they deal with these children while in school.

Reviewers' comments:

Reviewer's Responses to Questions

**Comments to the Author**

1. If the authors have adequately addressed your comments raised in a previous round of review and you feel that this manuscript is now acceptable for publication, you may indicate that here to bypass the “Comments to the Author” section, enter your conflict of interest statement in the “Confidential to Editor” section, and submit your "Accept" recommendation.

Reviewer #1: All comments have been addressed

Reviewer #2: All comments have been addressed

2. Is the manuscript technically sound, and do the data support the conclusions?

Reviewer #1: (No Response)

Reviewer #2: Yes

3. Has the statistical analysis been performed appropriately and rigorously? 

Reviewer #1: (No Response)

Reviewer #2: Yes

4. Have the authors made all data underlying the findings in their manuscript fully available?

Reviewer #1: (No Response)

Reviewer #2: Yes

5. Is the manuscript presented in an intelligible fashion and written in standard English?

Reviewer #1: (No Response)

Reviewer #2: Yes

6. Review Comments to the Author

Reviewer #1: (No Response)

Reviewer #2: No additional comments at this point. Am satisfied with the revisions and responses to the reviewer queries

7. PLOS authors have the option to publish the peer review history of their article (what does this mean?). If published, this will include your full peer review and any attached files.

Reviewer #1: **Yes:** Carlton Dampier MD

Reviewer #2: **Yes:** Conrad K Muzoora

---

## [Author Response · Author response to Decision Letter 2]

23 Apr 2026

MAKERERE UNIVERSITY

COLLEGE OF HEALTH SCIENCES

SCHOOL OF MEDICINE

DEPARTMENT OF PEDIATRICS AND CHILD HEALTH

23rd April 2026

Dr. Aloysius Gonzaga Mubuuke

Academic Editor

PLOS One

Dear Dr. Mubuuke,

On behalf of my co-authors, I am pleased to submit our responses to the reviewers’ comments for the manuscript titled “School absenteeism among children and adolescents aged 6–19 years with sickle cell disease in Uganda: a comparative cross-sectional study” (Manuscript ID: PONE-D-25-63605). We sincerely thank the Academic Editor and reviewers for their careful evaluation of our manuscript and for their valuable and constructive comments. We have carefully addressed each comment in detail below and revised the manuscript accordingly.

EDITOR COMMENTS

Comment 1: The paper has significantly improved and is benefit to the readership of PLOS ONE. The discussion has been strengthened. However, before the authors explain the strengths and limitations of the study towards the end of the discussion, they should include some key implications of the findings from this study first to clinicians who take care of these children with SCD and second, to the school administration authorities especially that they deal with these children while in school.

Response: We have revised the Discussion section to include a new paragraph on the practical implications of our findings before the strengths and limitations section. Specifically, we added implications for clinicians caring for children and adolescents with sickle cell disease (SCD), emphasizing routine screening for school absenteeism, early management of pain crises and hospitalizations, caregiver counseling, and psychosocial support. We also included implications for school administrators and education authorities, highlighting the need for flexible attendance policies, teacher sensitization, improved communication between schools and families, and academic support for learners with SCD.

Revised text: Discussion section, page 18-19, lines 341-360

“These findings have important implications for clinicians caring for children and adolescents with sickle cell disease. Routine clinical care should include screening for school absenteeism as part of holistic patient assessment, since frequent absence may signal worsening disease severity, psychosocial distress, or barriers to treatment adherence. Early identification and management of pain crises, infections, and other complications that lead to hospitalization may help reduce school disruption. Clinicians should also counsel caregivers on the importance of sustained school attendance and work with families to schedule routine clinic visits, where possible, outside critical school periods. Strengthening psychosocial support and referral systems for learners with repeated absenteeism may further improve both health and educational outcomes.

The findings also have important implications for school administrators and education authorities. School administrators should recognize sickle cell disease as a chronic health condition requiring flexible and supportive attendance policies rather than punitive responses to repeated absence. Improved communication between schools, caregivers, and healthcare providers may help schools better understand illness-related absenteeism and support timely academic catch-up. Teacher sensitization on the challenges faced by learners with SCD is important to reduce stigma, improve classroom support, and encourage retention in school. Where feasible, schools should strengthen school health programs and provide reasonable accommodations such as flexible assessment schedules, remedial learning support, and referral pathways for children experiencing frequent illness-related interruptions in learning.”

We hope we have satisfactorily addressed all comments from the Academic Editor and reviewers, and that the revised manuscript is now suitable for publication in PLOS ONE.

We are sincerely grateful for the valuable suggestions, which have helped improve the clarity, interpretability, and overall quality of our manuscript.

Sincerely,

Nattimba Juliane Peninah

Corresponding author

---

## [Editor Report · Decision Letter 2]

13 May 2026

School absenteeism among children and adolescents aged 6-19 years with sickle cell disease in Uganda: a comparative cross-sectional study

PONE-D-25-63605R2

Dear Dr. Nattimba,

We’re pleased to inform you that your manuscript has been judged scientifically suitable for publication and will be formally accepted for publication once it meets all outstanding technical requirements.

Kind regards,

Aloysius Gonzaga Mubuuke

Academic Editor

PLOS One

Additional Editor Comments (optional):

The paper reads well and will be of importance to the journal readership
---

## [Editor Report · Acceptance letter]

PONE-D-25-63605R2

PLOS One

Dear Dr. Nattimba,

I'm pleased to inform you that your manuscript has been deemed suitable for publication in PLOS One. Congratulations! Your manuscript is now being handed over to our production team.

Kind regards,

on behalf of

Dr. Aloysius Gonzaga Mubuuke

Academic Editor

PLOS One